# Biostimulant and Bioinsecticidal Effect of Coating Cotton Seeds with Endophytic *Beauveria bassiana* in Semi-Field Conditions

**DOI:** 10.3390/microorganisms11082050

**Published:** 2023-08-09

**Authors:** Spiridon Mantzoukas, Vasileios Papantzikos, Spiridoula Katsogiannou, Areti Papanikou, Charalampos Koukidis, Dimitrios Servis, Panagiotis Eliopoulos, George Patakioutas

**Affiliations:** 1Department of Agriculture, University of Ioannina, Arta Campus, 45100 Ioannina, Greece; b.papantzikos@uoi.gr (V.P.); agr1715765@uoi.gr (S.K.); agu16251@uoi.gr (A.P.); gpatakiu@uoi.gr (G.P.); 2BASF Hellas, 15125 Marousi, Greece; charalampos.koukidis@basf.com (C.K.); dimitris.servis@basf.com (D.S.); 3Laboratory of Plant Health Management, Department of Agrotechnology, University of Thessaly, 41500 Larissa, Greece

**Keywords:** endophytes, entomopathogens, aphids, biostimulant, bioinsecticidal, plant growth

## Abstract

Increasing commercial demands from the textile and food industries are putting strong pressure on the cultivation of cotton and its derivatives to produce high yields. At the same time, cotton has high nutrient and irrigation requirements and is highly susceptible to insect pests. Coating cotton seeds with beneficial fungi could address these problems. The aim of this study was to investigate the growth of cotton using (A) conventional seeds and (B) seeds coated with the entomopathogenic fungus *Beauveria bassiana* (Hypocreales: Cordycipitaceae). The experiment was conducted in a greenhouse of the Department of Agriculture of the University of Ioannina, in a completely randomized design. The growth characteristics of cotton plants were recorded weekly while the fresh weight and dry matter of the leaves, shoots and roots of the developed cotton plants were calculated at the end of the experiment. Weekly determination of total chlorophyll content (TCHL) was used as an indicator of plant robustness during the 80-day experiment. Many cotton growth parameters of treated plants, like number of leaves, shoots and apical buds, plant height, stem diameter, fresh and dried biomass and TCHL, were significantly higher than those of the untreated ones. Apart from plant growth, naturally occurring by *Aphis gossypii* (Hemiptera: Aphididae) infestation which also monitored for six weeks. A significantly lower aphid population was recorded for inoculated plants after the fifth week compared to the control. The overall evaluation revealed that *B. bassiana* coating treatments appear to have a significant biostimulatory and bioinsecticidal effect. Our results could represent responsive applications to the demands of intensive cotton growing conditions.

## 1. Introduction

Climate is a vital parameter for cotton cultivation [1], as severe drought has a negative impact resulting in poor cotton fiber quality [2], as well as lower yields [1,3]. The robustness of cotton roots is a crucial factor for the yield of quality lint, while soil conditions play a crucial role in the development and spread of the root system [4]. Therefore, the selection of the cultivation soil and the seed variety is very important. Another problem in cotton cultivation is insect pests [1,5] like aphids and thrips [6], which may cause severe yield losses. Another challenge for entomologists today is to develop eco-friendly solutions for cotton pest control, avoiding the drawbacks of chemical measures such as environmental pollution and insecticide resistance, while aiming to provide beneficial microbe-based insecticide solutions [7].

The problems in cotton cultivation require solutions to ensure the optimal production and quality of cotton derivatives. Fortifying cotton seeds with certain coatings that improve their quality and subsequent resistance to pest and/or drought stress may provide solutions to some of these issues. Recently, cotton seed coatings with organic acid mixtures such as citric acid, glutamate and proline have been proposed [8], with encouraging results for high yields, while coating seeds with entomopathogenic fungi (EPF) is of interest for aiding the adaptation of plants during their development [9] while helping to prevent diseases [10].

In recent years, the symbiosis between plants and entomopathogenic ascomycetes such as *B. bassiana* has become an interesting field of research, not only because of the endophytic behavior characteristic of arthropod pathogenic fungi, but also because of the positive influence that endophytic ascomycetes exert on plants by helping them to develop mechanisms to overcome stressful biotic factors, such as pests or fungal pathogens [11]. Endophytes also produce secondary metabolites that can potentially be used in agriculture [7,12] and improve plant host tolerance to abiotic stress, fungal diseases or pests [13], and in return plants provide a friendly environment allowing for beneficial fungi to obtain nutrients [14] and disperse. Fungal endophytes are defined as fungi that colonize internal plant tissues for all or part of the plant’s life, and this symbiosis has no adverse effects on plant growth [15].

Endophytes are microorganisms that spend part of their life in a nonparasitic association with plants [16]. There is ample evidence that many entomopathogenic fungi undergo an endophytic phase in various plant species. Typically, this endophytic relationship confers insecticidal or insect repellent properties to plants, a property that could be exploited for the development of environmentally friendly pest control applications. Fungal entomopathogens, such as *Beauveria bassiana* (Balsamo) Vuillemin (Hypocreales: Cordycipitaceae); *Metarhizium anisopliae* (Metschnikoff) Sorokin (Hypocreales: Clavicipitaceae); and *Isaria fumosorosea* (Wize) (Hypocreales: Clavicipitaceae), are important biocontrol agents of insect populations. It has been proposed that fungal endophytes could be a promising substitute for conventional insecticides and transgenic plants [17]. Moreover, the symbiosis of endophytes with the host plant is usually asymptomatic and causes the plant to respond differently to environmental changes [1,2,3,4].

The entomopathogenic fungus *B. bassiana* has been reported in the literature as an endophyte [11,18,19]. Its use does not present opposite effects on crops [20]. It shows a higher performance, as an inoculant, through vertical seed transmission in several crops, such as rice, *Oryza sativa* L. (Poales: Poaceae) [15]; sorghum, *Sorghum bicolor* (L.) Moench (Poales: Poaceae) [21]; broad bean, *Vicia faba* L. (Fabales: Fabaceae) [18]; Montrey pine, *Pinus radiata* D.Don (Pianles: Pinaceae) [19]; and opium poppy, *Papaver somniferum* L. (Ranunculales: Papaveraceae) [11]. Various methods have been proposed for artificial seed inoculation, such as seed coating and seed soaking (referred to as seed treatment or seedling inoculation) [22].

The use of the endophyte EPF *B. bassiana* as a seed inoculant can improve the metabolism of cotton plants, and it can also be used as a potential biostimulant [23]. This provides an additional approach for an Integrated Pest Management (IPM) program. The symbiotic relationship between the host and fungus–plant interactions can enhance plant robustness, as evidenced by growth characteristics such as total plant length and fresh and dry plant biomass [24].

When applied as an endophyte EPF in crops, the beneficial fungus *B. bassiana* promotes root development, thus enhancing nutrient uptake by the plant [25]. It presents entomopathogenic properties in cotton pests [26] in addition to exhibiting remarkable biostimulant action [27]. As a cotton seed inoculant, *B. bassiana* has been shown to have a biostimulant effect by suppressing the population of cotton bollworm *Helicoverpa armigera* (Hübner) (Lepidoptera: Noctuidae), on the one hand, and by transporting nutrients from soil to the roots on the other hand, thereby demonstrating potential for promoting growth [12].

In this work, an attempt was made to improve the quality of cultivated cotton seeds. Cotton seeds were coated with EPF *B. bassiana*, and their development and growth characteristics were monitored. Also, we evaluated the potential of endophyte cotton plants to become less suitable for *A. gossypii* nymphs.

## 2. Materials and Methods

### 2.1. Protocol for Coating Seeds with B. bassiana

Treatment solutions were prepared using dispensers and pipettes according to the desired treatment (125 mL of formulated *B. bassiana*/100 Kg cotton seed). In particular, the pipettes were used for the addition of the *B. bassiana* strain PPRI 5339 Velifer^®^ OD (BASF SE, Florham Park, NJ, USA), while the dispenser was used for the addition of the required amount of water. Then, the seed samples were separated using a precision balance (KERN PES 6200-2M). The final application of the coating to the seeds was carried out using the Wintersteiger Hege 11. First, the seed sample was placed in a special stainless-steel bucket. Stirring began, and finally, an appropriate amount of seed treatment solution was applied using the Eppendorf pipette. The entire application process took ~1 min/treatment/sample. The treated seeds (sample) were emptied into a special sample bag, which was left open until the seeds were completely dry. During the rotation of treatments, the equipment used to apply the treatment solution and the application bucket were thoroughly cleaned with ddH_2_O. The fungal spore concentration was zero in the control and 10^8^ spores/mL in the coated seeds. We used a Neubauer hemocytometer (TIEFE 0.100 mm 1/400 9 mm) to calculate the number of conidia (carried out in March 2022).

### 2.2. Germination Percentage

Experimental seeds (three replications of 30 seeds) were placed between moist germination paper (20 × 25 cm). Whatman paper 1 was placed on the Petri dish and moistened until thoroughly damp (in our case, ~2 mL of water was added). The papers were rolled and wrapped in a sheet of plastic to reduce evaporation, and the Petri dishes were placed in an incubator at 25 °C (PHC Europe/Sanyo/Panasonic Biomedical MLR-352-PE) in an upright position. Seeds were considered to have germinated when the radicle protruded by about ≥2 mm. The germination percentage was recorded every 24 h for 6 days. The procedure was performed under a laminar flow hood (Equip Vertical Air Laminar Flow Cabinet Clean Bench, Equip Mechanical Applications LTD, Piraeus, Greece). The germination percentage was calculated as follows:Germination Percentage = germinated seeds/total seeds × 100

After coating the cotton seeds with *B. bassiana*, the percentage of germination was calculated compared to the conventional seeds. After germination, seeds were transferred to 4 L pots containing a growth substrate composed of a mixture of peat and perlite in 1:1 ratio (*v*/*v*), fertilized with an N30-P10-K10 fertilizer and irrigated daily through a drip system (ARGOS electronics 2014) automatically controlled by a computer. Irrigation quantity and frequency were based on climatic data from temperature and humidity sensors. The Treatments were arranged in a complete randomized design with three replications. The experiment took place in the greenhouse of the Department of Agriculture of the University of Ioannina, Arta campus (39°07′15.7″ N 20°56′44.7″ E).

### 2.3. Assessment of Natural Infestation by Insects

Leaf sampling of experimental cotton plants was carried out from September to December 2022 every 7 days. Twenty cotton plants were sampled from each of the two plots (treated and control plants). During each sampling, twenty leaves of the same age and size were taken from the center of the plants of each plot (one leaf per plant). The leaves were always collected from the same plants. Collected leaves were put in paper bags and transferred to the lab for observation. Following this, the number of 3rd–4th-instar nymphs of *A. gossypii* per leaf was counted visually using a ×7 head lens (optiVISOR, LightCraft, London, UK). These developmental stages were preferred over adult and juvenile nymphs for counting as a population indicator given that this is more practical and reliable.

### 2.4. Plant Growth Parameters

Total plant length (cm), shoot diameter (mm), number of lateral shoots, number of internodes, total number of buds and number of leaves were monitored weekly and after 80 days. Leaf area (cm^2^), fresh weight of leaves, stems and roots (g) and their dry matter (g) were recorded. At the end of the experiment, shoots, leaves, roots and stems were separated, cleaned and cut into smaller pieces. Subsequently, they were oven-dried at 80 °C for 72 h. The fresh and dry mass of each plant part was recorded (n = 24).

### 2.5. Plant Colonization by the B. bassiana Endophyte

To verify the establishment of the endophyte in the plant tissues, randomized cotton leaves were taken with sterile scissors seven days after the treatment. Leaf samples were cut into 1 cm diameter and 0.5 cm thick discs in a laminar flow chamber. The samples were immersed in 96% ethanol solution for 1 min, in 6% sodium hypochlorite solution for 5 min and in 96% ethanol solution for 30 sec for sterilization [28]. After that, they were transferred to SDA substrate. Following this, samples on SDA were kept in incubators (25 ± 2 °C, 80% RH, darkness). After fourteen days, conidial germination was checked using an optical microscope (40×). The number of leaves which displayed fungal growth was calculated using the following formula: number of leaves with fungal growth/total number of samples [28,29].

### 2.6. Total Chlorophyll Content of Cotton Leaves

The total chlorophyll content (TCHL) of leaf tissues was monitored weekly and estimated using the linear correlation of the non-destructive method with the SPAD-502 instrument (Minolta Co., Ltd., Tokyo, Japan) and the conventional chemical analysis according to the protocol of Priya and Ghosh [30] in randomly selected cotton leaf samples (R^2^ = 0.91643) with some modifications: 0.04 g of cotton leaf tissue was crushed in a mortar and pestle using 10 mL of acetone as an extraction solvent. The extract was filtered through a Whatman No. 4 filter paper and the absorbance was measured in a Jasco-V630 UV-VIS spectrophotometer, using the equations described by Lichtenthaler and Buschmann [31]. The result was expressed in µg of TCHL of fresh leaf per cm^2^ of cotton leaf area (Appendix A).

### 2.7. Statistical Analysis

For growth measurements, seed germination and colonization percentage, one-way ANOVA was performed. Tukey’s post hoc test was used to compare means of treatments. All statistical analyses were conducted using SPSS v. 25 (IBM-SPSS Statistics, Armonk, NY, USA).

## 3. Results

### 3.1. Seed Germination

The results of this research show that cotton seeds coated with the *B. bassiana* strain PPRI 5339 recorded a statistically higher germination percentage compared to the control seeds (F = 9.809, df = 1, *p* = 0.001). The coated seed germination ranged from 13.3% (24 h) to 100% (144 h), while for uncoated (control) seeds, it ranged from 9.2% (24 h) to 96.2% (144 h) (Figure 1).

### 3.2. Reisolation of Entomopathogenic Fungi from G. hirsutum Leaves on SDA Substrate

Endophytic *B. bassiana* was successfully reisolated from treated cotton leaves (Figure 2 and Figure 3). Mycelium began to appear 3 days later and had developed completely after 9 days at 25 ± 2 °C and in 95% humidity (Figure 3). A decline in colonization was observed after 14 days (F = 6.156, df = 1, *p* = 0.001).

### 3.3. Natural Infestation of A. gossypii Aphids on Cotton Plants

All plants colonized with entomopathogenic fungi demonstrated a significant effect on the aphid population in planta (F = 6.417, df = 1, *p* < 0.001) The average number of aphids on the leaf was 1.2 ± 0.6 at 37 DATs (15 December 2022) for *B. bassiana*-inoculated plants and 11.3 ± 0.6 for the control plants (Figure 4).

### 3.4. Cotton Growth Parameters

Inoculation with *B. bassiana* caused an increase in the number of leaves (Figure 5) and the plant’s height (Figure 6) by the end of the experiment. Differences proved to be statistically significant for both parameters (plant height: F = 6.702, df = 1, *p* = 0.019, number of leaves: F = 3.759, df = 1, *p* = 0.021). A similar increase was recorded in the stem diameter (F = 4.111, df = 1, *p* < 0.001) (Figure 7A), the number of apical buds (F = 2.379, df = 1, *p* < 0.001) (Figure 7B), the number of internodes (F = 3.936, df = 1, *p* = *p* < 0.001) (Figure 7C), and shoot length (F = 3.911, df = 1, *p* < 0.001) (Figure 7D).

### 3.5. Cotton Fresh and Dry Mass

Fresh and dry mass measured at final harvest are summarized in Table 1. *B. bassiana* significantly increased the fresh (F = 4.198, df = 1, *p* = 0.001) and dry (F = 5.811, df = 1, *p* = 0.001) mass of the aboveground part of cotton plants. The aboveground-to-belowground biomass ratio showed a slight but significant improvement (F = 6.133, df = 1, *p* = 0.001).

### 3.6. Total Chlorophyll Content (TCHL) of Cotton Leaves

Chlorophyll concentration was increased in *B. bassiana*-inoculated plants after 30 days and remained above that of the control plants until the end of the experiment (F = 3.360, df = 1, *p* < 0.001) (Figure 8). The increase in TCHL was attributed to the endophytes’ effect on the leaves. The situation changed after natural infestation with *A. gossypii* after 37 days. TCHL rapidly decreased in control plants due the infestation and leaf maturation. On the other hand, we also recorded a decrease in TCHL in inoculated plants, but this did not occur as rapidly as in the control. On the last day of the experiment, the TCHL was 41.12 μg cm^−2^ for the control plants and 62.01 μg cm^−2^ for *B. bassiana*-inoculated plants.

## 4. Discussion

In the present study, the *B. bassiana* strain PPRI 5339 was successfully reisolated from cotton leaves grown from coated seeds, indicating that cotton is a suitable host for this particular *B. bassiana* endophyte. Seed coating also resulted in significantly increased germination and plant growth. Several parameters, like the number of leaves, shoots, internodes and apical buds, plant height, stem diameter, fresh and dry mass and TCHL, were increased due to the presence of the *B. bassiana* endophyte. Treated plants were also more tolerant to aphid infestation, recording significantly lower aphid populations compared to the control plants.

Many factors can influence the outcome of an experiment to establish a fungal entomopathogen as an endophyte. These factors include the crop species and the fungal entomopathogen isolate used, the concentration of the inoculum, the age of the plant during the inoculations and the inoculation methods. Our findings are in contrast with [32], who observed that banana growth was not significantly enhanced by *B. bassiana*, even at the highest inoculum rate. Contrarily, our results agree with relevant studies reporting that endophytic strains of *B. bassiana* significantly increased the growth of soybean [22], wheat [33], cotton [27,34,35,36], cassava [37], coffee [38] and tomato [35,36,39]. As for other fungal entomopathogens, treatment with *Metarhizium* accelerated the vegetative development of winter wheat [40] and the root development of switchgrass and haricot beans [41]. Also, inoculation with *M. anisopliae* promoted the growth of tomato, but not when the lowest inoculation rate was applied [42]. A study by Lopez and Sword [27] showed that inoculating cotton plants with *Purpureocillium lilacinum* (Thom) Samson (Hypocreales: Ophiocordycipitaceae) increased certain growth parameters in cotton plants. In a study by Greenfield et al. [37], cassava plants grew faster after being inoculated with *M. anisopliae*. Other studies have also reported successful manipulation of fungal endophytes in plants under greenhouse and field conditions, like pepper [28], melon [43], strawberry [43] and tomato [44].

As mentioned above, coating seeds with *B. bassiana* significantly improved seed germination and plant growth. Similarly, Espinoza et al., 2019 [45] reported that the growth of chili seeds inoculated with *B. bassiana* and *M. anisopliae* strains was almost 10% greater than that of uninoculated seeds. Also, inoculating corn plants with *B. bassiana* resulted in higher seed germination [22]. Contrarily, maize seed germination was not affected by *Metarhizium* treatment [40,46].

Contradictory results are not surprising in the case of endophytes, since the endophytic ability of this particular fungus is strongly connected with the species of the host plant and even its variety [47]. Differences in the endophytic action of fungal entomopathogens could also be attributed to the innate characteristics of the tested strain [38].

Our results evidence the same positive effect of *B. bassiana* as an endophyte on an *A. gossyppi* population. Inoculated cotton plants showed a lower insect population, which represents a unique phenomenon in relation to *B. bassiana*, an endophytic entomopathogenic fungus. Host plants colonized by fungal endophytes can have substantial effects on insects that interact with these plants [48,49]. Endophytic establishment of *B. bassiana* reduced the survival and fecundity of various arthropods, such as aphids in a range of plants [50], the fall armyworm [51], the European corn borer [52], the Mediterranean corn borer [28] and the aphids *Myzus persicae* and *A. gossyppi* [43,53]. Endophytic EPF may affect plants in a similar manner to plant pathogens and endophytes, altering plant quality and other traits and thus insect performance [23,54]. Insect performance depends on the quality of the food consumed by plant-feeding insects, and a higher level of nitrogen in plant tissues usually enhances insect development and growth [55]. Insect herbivores and other stresses can be prevented by endophytic fungi [29,56,57,58]. Aphids may invest in reproduction when mortality risks are high [59]. Many insect species have been shown to increase some biological processes when exposed to low levels of stress [60]. It is possible that the presence of *B. bassiana* or the subsequent increase in host plant defenses stressed the aphids, causing them to invest more energy in reproduction. Aphids on control plants had higher reproduction rates than aphids on *B. bassiana*-inoculated plants. As a result of the higher aphid population in the control plants, the chlorophyll concentration and leaf area were affected. It is possible that the endophyte reduced the plant’s response to aphid stress. Additionally, endophytes modify a plant’s chemical profile by altering phytosterol composition, and they also compete with insects for nutrients [13]. Among the positively colonized *B. bassiana* plants, the overall aphid population was lower at all times of the experiment; the reduction was also significant in other endophytic plants when compared to control plants. Endophytes produce secondary metabolites that are not necessarily the reason for the negative impact on aphids observed in our study.

A plant’s physiology and growth should not normally be affected by established endophytes [22]. Endophytes, however, can sometimes enhance host resistance to stressful environmental conditions [61], such as drought and nutrient deficiencies [62], or strengthen the host’s defenses against biotic threats [32,63,64,65]. As a result of endophytic entomopathogenic fungi, plants are more resistant to insect herbivory and biomass loss [66].

## 5. Conclusions

Our study demonstrated for the first time the positive effects of the endophytic entomopathogen *B. bassiana* strain PPRI 5339 on plant growth enhancement in cultivated cotton. In addition to positive effects on plant growth, we also observed negative effects of the survival and development of a key herbivorous insect pest, *A. gossypii*. Importantly, although these effects were shown in semi-field trials, we have previously shown that the targeted manipulation of fungal endophyte strains in cotton can be achieved under field conditions using a simple seed inoculation protocol with the *B. bassiana* strain PPRI 5339. Understanding the environmental impacts on *B. bassiana*’s stability and colonization behavior through semi-field research can facilitate the design and establishment of more effective insect pest management strategies.

## Figures and Tables

**Figure 1 microorganisms-11-02050-f001:**
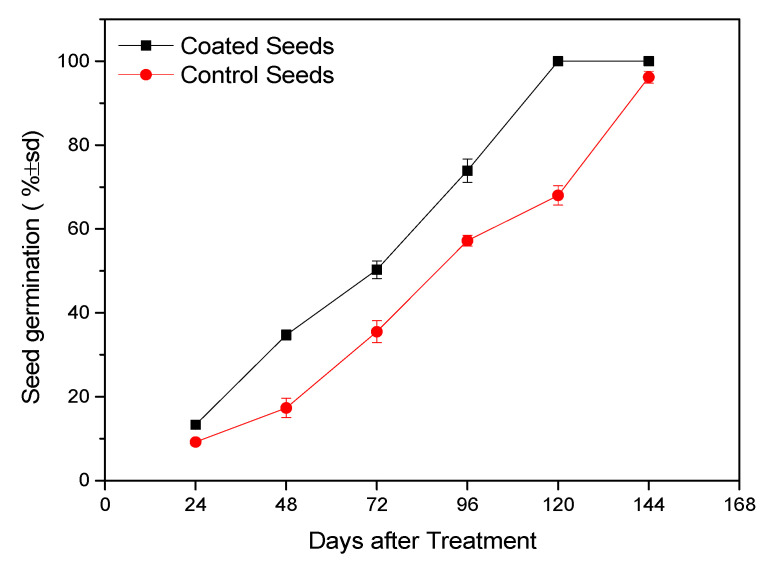
Germination of treated (coated) and untreated (control) cotton seeds.

**Figure 2 microorganisms-11-02050-f002:**
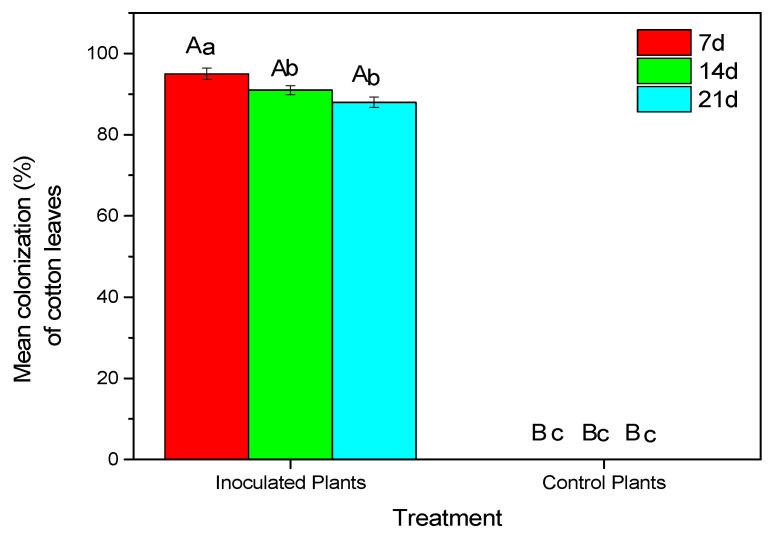
Mean (±sd; n = 24) colonization of cotton leaf parts by *B. bassiana* at 7 days, 14 days and 21 days after treatment (DATs). Mean values of the same treatment with the same small letter are not significantly different; mean values of the same DAT with the same capital letter are not significantly different (Tukey test: *p* < 0.05).

**Figure 3 microorganisms-11-02050-f003:**
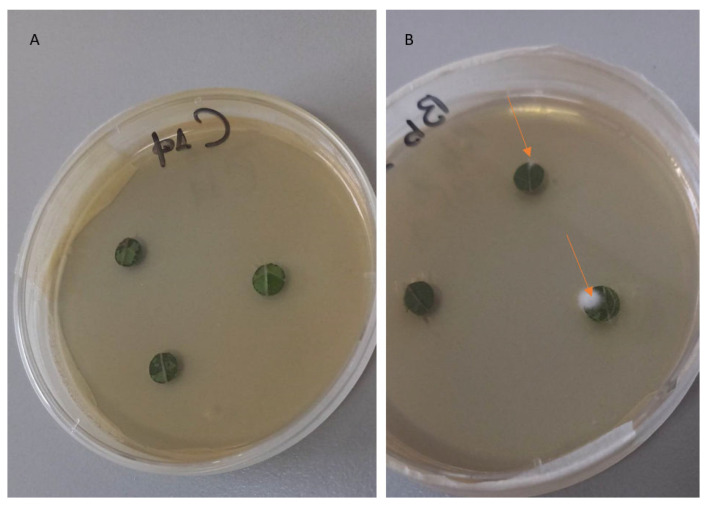
Successful reisolations after 3 days on SDA, when mycelium began to appear. (**A**) Control plant, (**B**) *B. bassiana*-inoculated plant. The orange arrows are showing the *B. bassiana* mycelium on cotton leaves.

**Figure 4 microorganisms-11-02050-f004:**
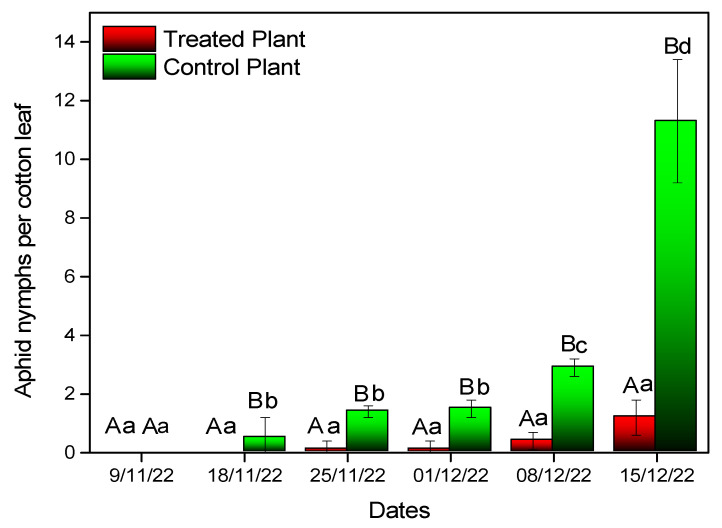
*A. gossypii* aphid population on the leaves of cotton plants after treatment with entomopathogenic fungi. Mean values of the same treatment with the same small letter are not significantly different; mean values of the same dates with the same capital letter are not significantly different (Tukey test: *p* < 0.05).

**Figure 5 microorganisms-11-02050-f005:**
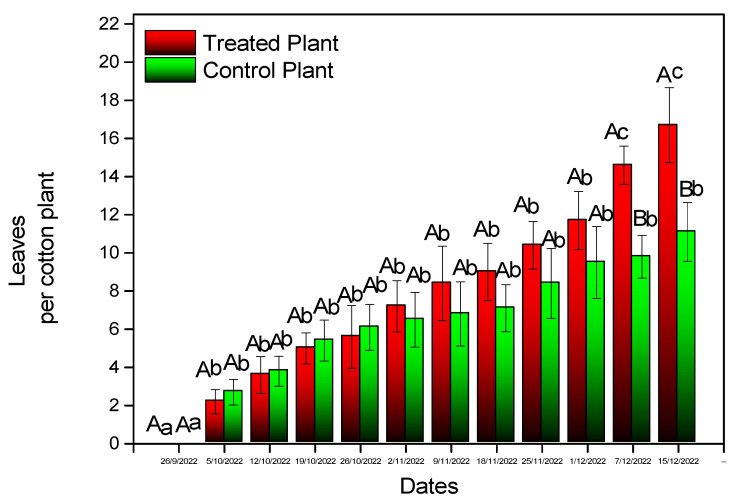
Average number of leaves of experimental cotton plants. Mean values of the same treatment with the same small letter are not significantly different; mean values of the same dates with the same capital letter are not significantly different (Tukey test: *p* < 0.05).

**Figure 6 microorganisms-11-02050-f006:**
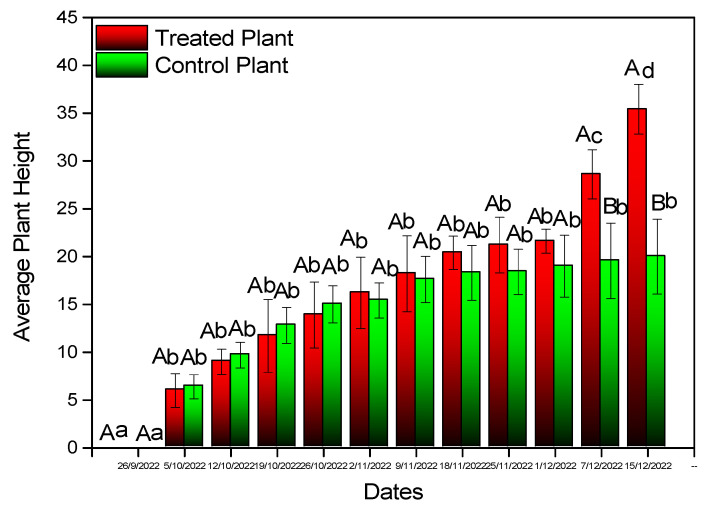
Average height of experimental cotton plants. Mean values of the same treatment with the same small letter are not significantly different; mean values of the same dates with the same capital letter are not significantly different (Tukey test: *p* < 0.05).

**Figure 7 microorganisms-11-02050-f007:**
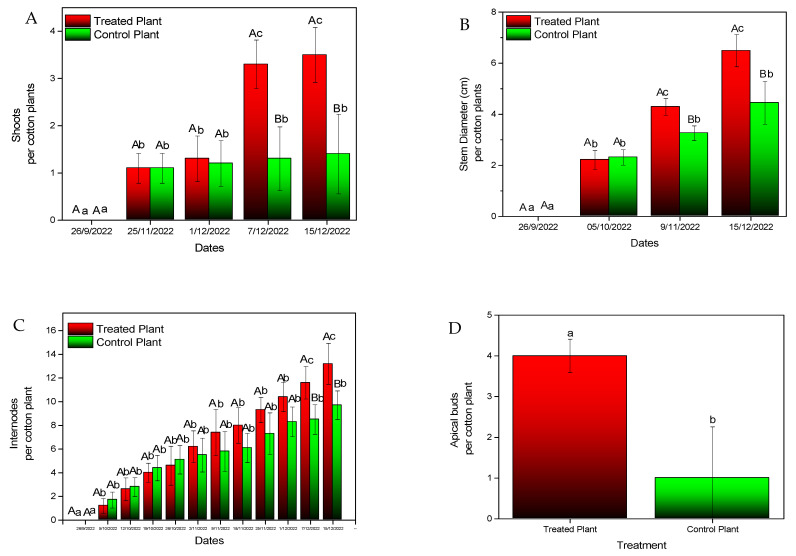
Growth parameters of experimental cotton plants. (**A**) Shoots, (**B**) stem diameter, (**C**) internodes and (**D**) apical buds. Mean values of the same treatment with the same small letter are not significantly different; mean values of the same dates with the same capital letter are not significantly different (Tukey test: *p* < 0.05).

**Figure 8 microorganisms-11-02050-f008:**
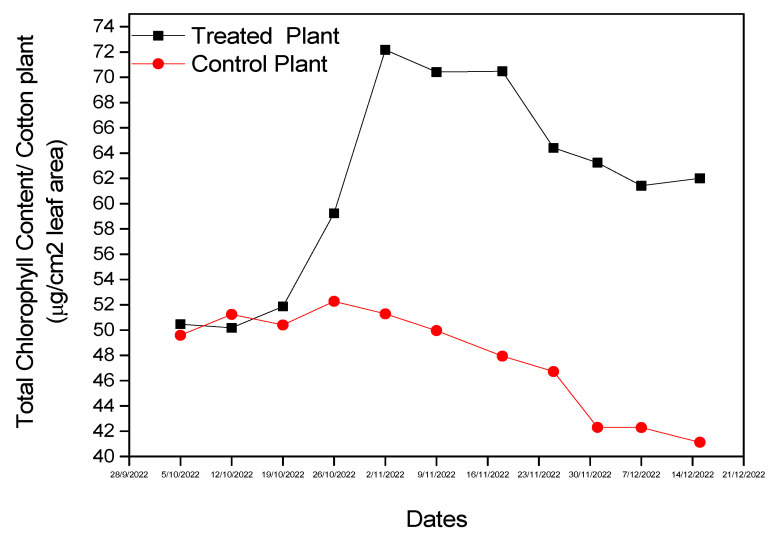
Chlorophyll concentration of experimental cotton plants.

**Table 1 microorganisms-11-02050-t001:** Cotton growth parameters of experimental cotton plants. Means of the same row followed by the same letter do not differ significantly.

Growth Parameter	Control	Treated
Aboveground fresh weight, A (g)	8.12 ± 1.14 a	20.6 ± 2.39 b
Belowground fresh weight, B (g)	2.58 ± 0.93 a	4.69 ± 0.64 b
Ratio A/B	3.14	4.39
Total fresh biomass (g)	10.70 ± 1.46 a	25.29 ± 3.73 b
Aboveground dry weight, A (g)	1.41 ± 1.56 a	4.89 ± 1.32 b
Belowground dry weight, B (g)	0.28 ± 0.15 a	0.60 ± 0.09 b
Ratio A/B	5.03	8.15
Total dry biomass (g)	1.69 ± 1.12 a	5.48 ± 1.84 b

## Data Availability

Not applicable.

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
