# Peer review of "Biostimulant and Bioinsecticidal Effect of Coating Cotton Seeds with Endophytic Beauveria bassiana in Semi-Field Conditions"

_microorganisms, 2023, doi:10.3390/microorganisms11082050_

Round 1

Reviewer 1 Report

Dear editor and authors,

Thank you for the opportunity to review this paper. I have thoroughly read the manuscript and provided minor comments that need to be addressed.

Overall, I believe the paper is well-written, and the experiment is valid and properly designed and analyzed. However, I suggest clarifying the results by improving the figures (axis labels are confusing).

Furthermore, I recommend that the authors reorganize the discussion section. The obtained results in this research are valuable and interesting, but they are challenging to understand in the current form of the discussion. My suggestion is to first explain the most significant findings of this experiment and then compare them with existing research.

There are some minor technical errors that will likely be corrected during the publication process.

Nevertheless, the paper is intriguing, and with the proposed revisions, I support its publication in the journal.

Best regards

Author Response

Thank you for the opportunity to review this paper. I have thoroughly read the manuscript and provided minor comments that need to be addressed. Overall, I believe the paper is well-written, and the experiment is valid and properly designed and analyzed.

Thank you very much for the kind comments.

However, I suggest clarifying the results by improving the figures (axis labels are confusing).

Axis labels of all Figures have been rephrased or corrected.

Furthermore, I recommend that the authors reorganize the discussion section. The obtained results in this research are valuable and interesting, but they are challenging to understand in the current form of the discussion. My suggestion is to first explain the most significant findings of this experiment and then compare them with existing research.

Thank you for the comment. The whole discussion has been re-organized as suggested. The first paragraph summarizes our main conclusions. Subsequently, we discuss the results on plant growth, seed germination and the effect on aphids. As you will see, sentences and paragraphs have been relocated to avoid unnecessary repetition and to provide a more structured flow in the text. We also removed some bibliographies that were not absolutely necessary.

There are some minor technical errors that will likely be corrected during the publication process. Nevertheless, the paper is intriguing, and with the proposed revisions, I support its publication in the journal.

Thank you very much for the kind comments.

Reviewer 2 Report

The authors present interesting results about the effect of the entomopathogenic fungus Beauveria bassiana strain PPRI 5339 on plant growth enhancement in cultivated cotton as well as the effects on the survival and development of a critical herbivorous insect pest, Aphis gossypii.

The present work was organized logically as a limitedly focused study. The results are well presented. However, I have some points that need to be addressed.

1. Line 128: “After coating the cotton seeds with B. bassiana,”. More details are required for this method.

2. Line 137: This method requires further elaboration.

3. Figure 1. It would be better to add this Figure to supplementary files.

The English language of the manuscript needs to be revised in places.

Author Response

  1. Line 128: “After coating the cotton seeds with B. bassiana,”. More details are required for this method.

The coating procedure is described analytically in the chapter 2.1.

  1. Line 137: This method requires further elaboration.

We added more details about the process we followed (highlighted with green).

  1. Figure 1. It would be better to add this Figure to supplementary files.

Figure 1 has been transferred to supplementary files (Figure S1). All other figures have been renamed accordingly.

Comments on the Quality of English Language. The English language of the manuscript needs to be revised in places.

Thank you for the remark. We did our best with the help of a British collaborator to improve language and grammar. We again checked the whole text point by point and made appropriate changes (highlighted with green).